# Towards High-Fidelity CAD Generation via LLM-Driven Program Generation and Text-Based B-Rep Primitive Grounding

**Jiahao Li**[1]  **Qingwang Zhang**[1]  **Qiuyu Chen**[2]  **Guozhan Qiu**[1]  **Yunzhong Lou**[1]  **Xiangdong Zhou**[1]

## Abstract

The field of Computer-Aided Design (CAD) generation has made significant progress in recent years. Existing methods typically fall into two separate categories: parametric CAD modeling and direct boundary representation (B-Rep) synthesis. In modern feature-based CAD systems, parametric modeling and B-Rep are inherently intertwined, as advanced parametric operations (e.g., *fillet* and *chamfer*) require explicit selection of B-Rep geometric primitives, and the B-Rep itself is derived from parametric operations. Consequently, this paradigm gap remains a critical factor limiting AI-driven CAD modeling for complex industrial product design. This paper presents *FutureCAD*, a novel text-to-CAD framework that leverages large language models (LLMs) and a B-Rep grounding transformer (*BRepGround*) for high-fidelity CAD generation. Our method generates executable CadQuery scripts, and introduces a text-based query mechanism that enables the LLM to specify geometric selections via natural language, which *BRepGround* then grounds to the target primitives. To train our framework, we construct a new dataset comprising real-world CAD models. For the LLM, we apply supervised fine-tuning (SFT) to establish fundamental CAD generation capabilities, followed by reinforcement learning (RL) to improve generalization. Experiments show that *FutureCAD* achieves state-of-the-art CAD generation performance. Code and dataset are available at https://github.com/JohanStackk/FutureCAD.

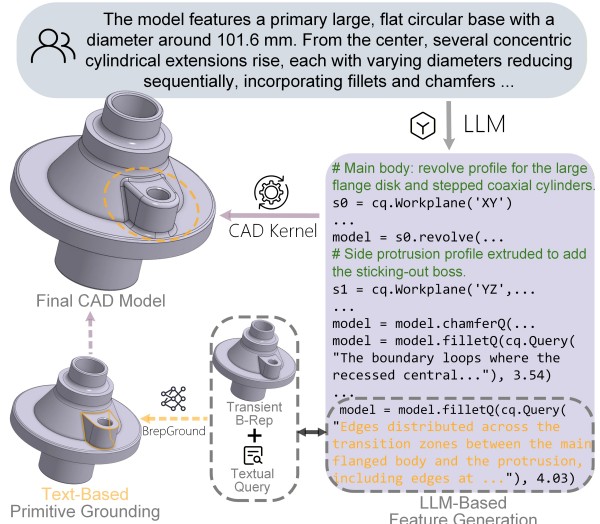

*Figure 1.* Given a textual description, *FutureCAD* synergizes LLM-driven program generation and text-based B-Rep primitive grounding to support feature-based CAD modeling, enabling high-fidelity CAD model generation.

## 1. Introduction

Computer-Aided Design (CAD) is fundamental to modern engineering and manufacturing, yet remains costly to master due to its steep learning curve (Cherng et al., 1998; Robertson & Allen, 2002). Consequently, AI-driven CAD generation has attracted growing attention and achieved notable progress in recent years (Daareyni et al., 2025). These methods typically fall into two distinct paradigms, namely, parametric modeling that represents CAD models as sequences of parametric operations and Boundary Representation (B-Rep) synthesis that explicitly models geometric and topological structures (Li et al., 2025b; Xu et al., 2025).

Parametric modeling approaches generate CAD models as procedural sequences of parametric operations, offering interpretability and editability. However, they fail to support advanced operations such as *fillet* and *chamfer*, which rely on B-Rep primitives that are absent from the parametric design history, limiting their applicability in realistic design scenarios. Conversely, B-Rep synthesis methods directly model geometry and topology, alleviating the aforementioned limitations, but sacrifice parametric structure and

---
[1]Fudan University [2]Shanghai Jiao Tong University. Correspondence to: Xiangdong Zhou <xdzhou@fudan.edu.cn>.

*Proceedings of the 43rd International Conference on Machine Learning*, Seoul, South Korea. PMLR 306, 2026. Copyright 2026 by the author(s).

design intent, making the resulting models difficult to reuse.

In modern feature-based CAD systems, parametric modeling and B-Rep are not two independent alternatives but are inherently intertwined (Shahin, 2008). The B-Rep is incrementally constructed during feature execution, and advanced operations operate by explicitly selecting primitives from the transient B-Rep at the time of execution. This tight coupling forms the basis for achieving complex and precise industrial designs. Nevertheless, existing CAD generation methodologies treat parametric modeling and B-Rep modeling as separate modeling problems (Heidari & Iosifidis, 2025). As a result, the fragmentation between these two paradigms constitutes a fundamental bottleneck that significantly limits the advancement of AI-driven CAD generation.

In this paper, we present *FutureCAD*, a general paradigm that integrates large language models (LLMs) with a boundary representation grounding transformer (*BRepGround*) to generate high-fidelity CAD models. *FutureCAD* employs an LLM to directly generate executable CadQuery scripts (CadQuery contributors), in which, when references to B-Rep primitives are required, the LLM specifies the target geometric entities by generating natural language queries. During program execution, *BRepGround* takes the transient B-Rep as input and grounds the textual query to the corresponding primitives. This design enables *FutureCAD* to seamlessly integrate with standard CadQuery workflows, while naturally aligning the LLM's strengths in code generation with its inherent proficiency in natural language querying, as illustrated in Figure 1.

To train our framework, we construct a new dataset comprising over 140k diverse and complex real-world CAD models that reflect realistic feature-based design practices. Each model is annotated with geometric descriptions for LLM training and textual queries for training *BRepGround* on primitive grounding. Subsequently, we convert these models into executable CadQuery programs, which serve as the final CAD representation used throughout our framework. We also augment the dataset by rewriting the CadQuery code to incorporate a richer set of native CadQuery APIs, thereby enhancing programmatic diversity. For the LLM, we adopt a two-stage training strategy: supervised fine-tuning (SFT) to equip the model with fundamental CAD generation and comprehension capabilities, followed by reinforcement learning (RL) to improve generalization and the validity of generated programs. Extensive experiments demonstrate that FutureCAD enables high-fidelity CAD generation, achieving state-of-the-art performance on both in-distribution and out-of-distribution benchmarks.

In summary, our contributions are as follows:

- We introduce *FutureCAD* as the first general text-to-CAD paradigm that unifies LLM-based program generation with text-based B-Rep primitive grounding, enabling practical feature-based CAD modeling.
- We construct a new dataset of over 140k real-world CAD models featuring rich parametric operations (e.g., *fillet*, *chamfer*, and *shell*), detailed geometric annotations, and CadQuery code representations.
- We introduce a text-based B-Rep grounding task and propose *BRepGround*, a transformer-based model that grounds textual queries to target geometric primitives.
- We demonstrate that our framework with a two-stage training strategy combining SFT and RL achieves state-of-the-art performance in high-fidelity CAD generation while maintaining a low invalid rate.

## 2. Related Work

**Parametric CAD Modeling** aims to generate editable construction sequences. Learning-based approaches typically model parametric sequence in a latent space (Wu et al., 2021), while structured representations have been introduced to encode design variations and enable controlled generation (Xu et al., 2022; 2023). Several works recover CAD construction sequences directly from point clouds (Khan et al., 2024a), while diffusion-based models explore alternative generative formulations (Ma et al., 2024). Transformer-based text-to-CAD generation has been explored via contrastive latent-space alignment (Li et al., 2024) and large-scale end-to-end autoregressive training (Khan et al., 2024b). Several recent efforts have explored learning-based CAD modeling with explicit support for advanced features via primitive selection or identification (Fan et al., 2025; Liu et al., 2025b); however, these methods rely on heuristic, rule-based matching, rather than explicit semantic modeling, which limits their generalization and scalability. Nevertheless, those approaches exhibit limited generalization, struggle to generate complex CAD models, and provide limited support for downstream tasks such as editing.

**LLM-Based CAD Generation** typically formulates CAD generation as structured sequence or code generation tasks (Zhang et al., 2025; Li et al., 2025b), by leveraging the strong generative priors of LLMs. Prior work predominantly adopts supervised learning on paired data, with conditioning signals spanning images, point clouds, and multi-modal inputs (Wang et al., 2025b; Rukhovich et al., 2025; Xu et al., 2024a). Recent approaches further incorporate feedback from geometric execution and visual evaluation to improve validity and geometric fidelity, including alternating visual-feedback training and reinforcement learning (RL)-based policy optimization (Wang et al., 2025a; Guan et al., 2025; Kolodiazhnyi et al., 2025; Li et al., 2026). In parallel, training-free and tool-augmented paradigms leverage self-refinement or closed-loop planning to reduce reliance on fine-tuning (Li et al., 2025c; Mallis et al., 2025). However,

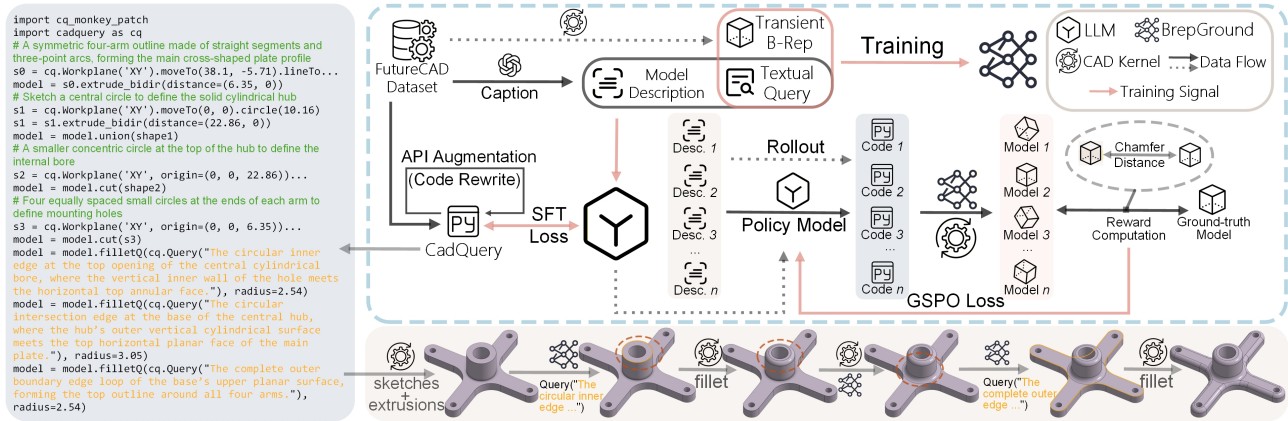

*Figure 2.* **Overview of the *FutureCAD* framework. *Left:*** An example CadQuery program with text-based queries (highlighted in orange) for B-Rep primitive selection. ***Top:*** The training pipeline includes *BRepGround* training for primitive grounding, and LLM training with supervised fine-tuning (SFT) followed by reinforcement learning (RL) with GSPO using Chamfer Distance-based rewards. ***Bottom:*** Illustration of the CAD modeling process: the LLM generates parametric features executed by the CAD kernel, and when operations require primitive references, the LLM produces textual queries for *BRepGround* to resolve, with the results enabling the kernel to proceed.

realistic feature execution demands more than producing a sequence: it also requires mapping intent to specific entities in the intermediate B-Rep, a requirement that is not naturally addressed by existing methods, where the modeling process is confined to a purely parametric representation. *FutureCAD* bridges this gap through semantic B-Rep primitive grounding alongside feature-based program generation.

**Direct B-Rep Modeling** targets synthesizing CAD models in their native boundary representation (B-Rep). *SolidGen* predicts explicit primitives with structured references to enforce entity consistency (Jayaraman et al., 2023). Diffusion or autoregressive formulations include hierarchical latents with merge-based topology recovery that improve realism and geometric complexity (Xu et al., 2024b), as well as compact encodings of NURBS parameters that reduce UV-grid overhead while preserving fidelity (Fan et al., 2024). Recent efforts emphasize structural correctness by explicitly separating topology sampling from subsequent geometry synthesis (Li et al., 2025a), whereas *AutoBrep* unifies geometry and topology into a single token stream with reference tokens and graph-aware ordering (Xu et al., 2025). Another direction strengthens geometry and topology coupling in latent space to improve validity (Liu et al., 2025c), and enables language-guided edits without modeling history (Liu et al., 2025a). However, direct B-Rep generators fail to preserve parametric structure and design intent. Consequently, the generated shapes are difficult to reuse and edit, and frequently yield invalid models.

## 3. Problem Definition

In this section, we formalize the modern feature-based modeling workflow in Sec. 3.1, followed by an overview of how our framework aligns with it in Sec. 3.2.

### 3.1. Feature-Based Design Workflow

Modern feature-based CAD models are constructed as an ordered sequence of features, where each feature instantiates a specific operation type with its associated parameters, and may additionally require explicit references to B-Rep primitives (e.g., faces or edges) as operands (Safdar et al., 2020). We formalize a CAD model as a feature sequence

$$\mathcal{F} = (f_1, f_2, \ldots, f_T), \tag{1}$$

where each feature is represented by a tuple

$$f_i = (t_i, \theta_i, \pi_i). \tag{2}$$

Here $t_i \in \mathcal{T}$ denotes the operation type (e.g., *sketch*, *revolve* and *fillet*), $\theta_i$ collects the operation-specific continuous and discrete parameters (e.g., distances, radii, angles, boolean modes), and $\pi_i$ denotes an optional set of referenced geometric primitives required to instantiate the operation. In particular, operations such as *fillet*, *chamfer*, and *shell* typically require $\pi_i \neq \emptyset$, where $\pi_i$ contains selected B-Rep faces or edges to be modified.

A CAD kernel executes $\mathcal{F}$ sequentially and maintains an evolving B-Rep state. We denote by $B_i$ the B-Rep after executing the first $i$ features, with an initial state $B_0$ that may include default geometry such as reference planes and the origin. The execution dynamics can be written as

$$B_i = \Phi(B_{i-1}, f_i), \qquad i = 1, \ldots, T, \tag{3}$$

where $\Phi$ is the CAD kernel transition operator that applies feature $f_i$ to the transient B-Rep state $B_{i-1}$.

Crucially, when a feature requires explicit primitive references, its operand set must be resolved from the *current* B-Rep state. Let $\mathcal{P}(B)$ denote the set of available primitives in a B-Rep $B$. Then the referenced primitives satisfy

$\pi_i \subseteq \mathcal{P}(B_{i-1})$, i.e., primitive references for step $i$ are selected from the B-Rep produced by all preceding features.

## 3.2. Overview of FutureCAD

Given a natural language description $x$ as input, our goal is to generate an executable CadQuery program that encodes a sequence of $T$ parametric features, whose execution produces a final B-Rep $B_T$ representing the target CAD model. The overall framework is illustrated in Figure 2. Specifically, *FutureCAD* first uses an LLM to synthesize an executable CadQuery program $P$ conditioned on $x$, whose semantics encodes the feature sequence $\mathcal{F}$:

$$P = \text{LLM}(x), \qquad [\![P]\!] = \mathcal{F}, \qquad (4)$$

where the generated program specifies the feature types $\{t_i\}_{i=1}^T$ and parameters $\{\theta_i\}_{i=1}^T$, and embeds a textual query $q_i$ whenever feature execution requires primitive references.

*FutureCAD* couples this program-level feature generation with a B-Rep grounding module, termed *BRepGround*, to resolve primitive references. During program execution, for any feature $f_i$ associated with a query $q_i$, *BRepGround* takes the transient B-Rep state $B_{i-1}$ and grounds the query to the corresponding primitive references:

$$\pi_i = \text{BRepGround}(B_{i-1}, q_i), \quad \text{with } \pi_i \subseteq \mathcal{P}(B_{i-1}). \qquad (5)$$

The resolved primitives $\pi_i$ are then supplied back to the CAD kernel to execute $f_i$, yielding the B-Rep transition in Eq. (3). In this manner, *FutureCAD* tightly couples language-driven feature generation with B-Rep primitive grounding. Details of *BRepGround* and LLM-based feature generation are presented in Sec. 4 and Sec. 5, respectively.

## 4. BRepGround Architecture

The *BRepGround* architecture encodes textual queries and B-Rep primitives into embedding spaces and fuses them via a transformer-based module, as illustrated in Figure 3.

**Text Encoder** employs a pretrained BERT (Devlin et al., 2019) model to obtain embeddings of the natural language query $q$,

$$\mathbf{H}^{\text{text}} = \text{BERT}(q), \qquad (6)$$

where $\mathbf{H}^{\text{text}} \in \mathbb{R}^{L \times d}$ denotes the contextualized token embeddings of the query, $L$ is the query length, and $d$ is the hidden dimension of the text encoder.

**B-Rep Encoder** employs UV-Net (Jayaraman et al., 2021) to extract embeddings for faces and edges. Each face is represented by a 2D UV-grid sampled on its parametric surface, while each edge is represented by a 1D UV-grid sampled along its parametric curve. Edges are represented bidirectionally and the geometric grids are augmented with auxiliary geometric attributes.

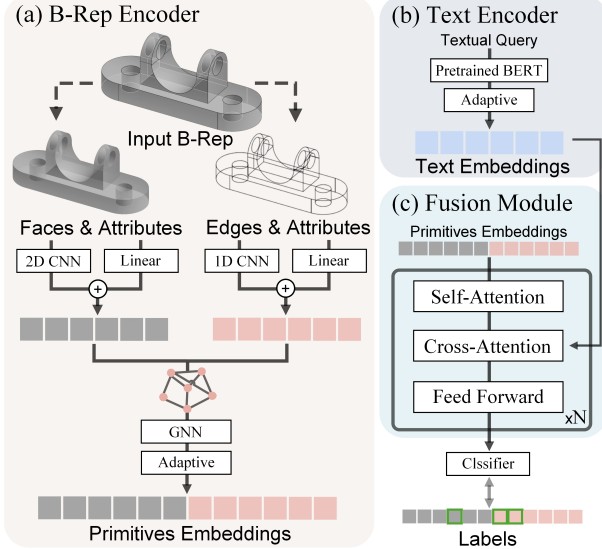

*Figure 3.* **Architecture of *BRepGround*. (a) The B-Rep encoder** extracts face and edge features and produces primitive embeddings via a GNN and adaptive layer. **(b) The text encoder** processes the query using pretrained BERT. **(c) The fusion module** fuses primitive and text embeddings through self-attention and cross-attention, followed by a classifier that predicts target primitives.

Formally, given a B-Rep $B$, UV-Net produces geometric embeddings for faces and edges. The resulting representations are defined as $\mathbf{H}^{\text{face}} = [\text{UV}_{\text{surf}}(B); \text{Linear}(\mathbf{a}^{\text{face}})]$ and $\mathbf{H}^{\text{edge}} = [\text{UV}_{\text{curve}}(B); \text{Linear}(\mathbf{a}^{\text{edge}})]$, where $\mathbf{a}^{\text{face}}$ and $\mathbf{a}^{\text{edge}}$ denote raw geometric attributes at the face and edge levels. The face and edge attributes capture geometric and topological properties, and the resulting embeddings concatenate UV-Net geometric features with attribute information. These embeddings are then propagated over the face-adjacency graph using a graph encoder (Xu et al., 2019):

$$\left(\mathbf{Z}^{\text{face}}, \mathbf{Z}^{\text{edge}}\right) = \text{GNN}\left(\mathbf{H}^{\text{face}}, \mathbf{H}^{\text{edge}}\right), \qquad (7)$$

where GNN denotes a graph neural network that performs message passing between adjacent faces and edges to produce context-aware face and edge embeddings. Bidirectional edge embeddings are averaged after graph encoding to form undirected edge embeddings for subsequent fusion.

**Adaptive Layer.** To align embeddings from different modalities, we adopt an Adaptive Layer (Khan et al., 2024b) that is applied after both the text encoder and the B-Rep encoder. The Adaptive Layer consists of a multi-head self-attention module (Vaswani et al., 2017) followed by a position-wise feed-forward network, each wrapped with residual connections and layer normalization (Zhang & Sennrich, 2019).

For the B-Rep modality, face and edge embeddings are jointly processed by the Adaptive Layer with an additional segment embedding to distinguish primitive types. Specifically, face embeddings and edge embeddings are first con-

catenated into a single sequence, and a learnable segment embedding is added to indicate whether a token corresponds to a face or an edge:

$$\mathbf{H}^{\text{brep}} = \text{Adaptive}\left(\left[\mathbf{Z}^{\text{face}};\ \mathbf{Z}^{\text{edge}}\right] + \mathbf{E}^{\text{seg}}\right), \qquad (8)$$

where $\mathbf{E}^{\text{seg}}$ denotes a segment embedding that assigns different embeddings to face and edge tokens.

**Fusion Module.** To fuse language and geometric representations, we adopt a transformer-based fusion module that operates on B-Rep tokens conditioned on text embeddings. The fusion module takes the output of the Adaptive Layer as geometric tokens and performs iterative self- and cross-attention with the text embeddings.

Formally, let $\mathbf{H}^{\text{brep}} \in \mathbb{R}^{L_{\text{uv}} \times d}$ denote the sequence of B-Rep tokens (faces and edges), and let $\mathbf{H}^{\text{text}} \in \mathbb{R}^{L_{\text{text}} \times d}$ denote the text token embeddings. At each fusion layer, the B-Rep tokens are first updated by self-attention, followed by cross-attention where B-Rep tokens attend to text tokens, and a position-wise feed-forward network:

$$\mathbf{H}^{\text{brep}} \leftarrow \text{FFN}\left(\text{CA}\left(\text{SA}\left(\mathbf{H}^{\text{brep}}\right), \mathbf{H}^{\text{text}}\right)\right), \qquad (9)$$

where SA denotes multi-head self-attention over B-Rep tokens, CA denotes multi-head cross-attention with B-Rep tokens as queries and text tokens as keys and values, and FFN denotes a position-wise feed-forward network. All sub-layers are wrapped with residual connections and layer normalization.

**Training.** The fused B-Rep token embeddings are passed through a prediction head that predicts a scalar score for each B-Rep token:

$$\mathbf{s} = \text{MLP}\left(\mathbf{H}^{\text{brep}}\right), \qquad (10)$$

where $\mathbf{H}^{\text{brep}} \in \mathbb{R}^{L_{\text{uv}} \times d}$ denotes the fused B-Rep token embeddings, and $\mathbf{s} \in \mathbb{R}^{L_{\text{uv}}}$ denotes the predicted logits for all face and edge tokens.

Training supervision is provided as binary labels over B-Rep tokens. The model is trained with a weighted binary cross-entropy loss:

$$\mathcal{L}_{\text{ref}} = \frac{1}{N} \sum_{i=1}^{N} \text{BCE}\left(s_i,\ y_i;\ w^+\right), \qquad (11)$$

where $s_i$ and $y_i$ denote the predicted logit and ground-truth label for token $i$, respectively, and $w^+$ denotes the positive class weight used to address class imbalance.

Training samples are constructed from feature operations with non-empty primitive references ($\pi_i \neq \emptyset$) in our dataset, pairing the pre-operation B-Rep state with a textual description obtained using Claude-Sonnet-4.5 (Anthropic, 2025) that refers to target primitives $\pi_i$.

## 5. LLM-Based CAD Program Generation

Large Language Models (LLMs) have demonstrated strong capabilities in reasoning and code generation (Gao et al., 2023; Wong et al., 2023), and are inherently proficient in understanding and producing natural language. This makes them well suited for our design, which requires the generation of both programs and textual queries. Motivated by this, we adopt an LLM to generate executable CadQuery programs, as CadQuery provides rich feature-based operations through a high-level and interpretable modeling interface.

### 5.1. Supervised Fine-Tuning Stage

The supervised fine-tuning (SFT) stage aims to equip LLMs with basic capabilities for CAD program generation and understanding by training them to generate executable CadQuery programs from natural language descriptions.

**Data Construction.** We first convert CAD models in our dataset into CadQuery programs $P$. To support natural language queries in CAD programs, we extend the CadQuery interface at runtime to resolve the referenced B-Rep primitives during execution, e.g., *filletQ(cq.Query("All circular holes on the surface."), radius=0.25)*. This design allows programs to specify feature operands through semantic descriptions of target primitives. Additionally, we extend the CadQuery interface to support parameterizations required by our dataset that are not exposed by its native API [1].

**Textual description generation.** Existing approaches typically rely on rendered images, which often fail to capture parametric information (Li et al., 2025b). Incorporating CAD sequences can address this issue but tends to produce overly detailed descriptions and cause information leakage (Khan et al., 2024b). Instead, we generate two levels of descriptions by jointly leveraging rendered images and CadQuery programs: an abstract description $x_a$ that summarizes overall shape and structure, and a detailed description $x_d$ that captures finer-grained geometric and scale information.

By leveraging LLMs' intrinsic code understanding and the semantic and geometric cues encoded in CAD programs, we obtain accurate yet non-redundant descriptions. An example of a detailed description in our dataset is: *"The hexagonal top has a span of approximately 40 units in width and 23.09 units in height, extruded upwards by 15 units. Each of its six edges is filleted with a radius of 2.5 units, softening the corners. The cylindrical pedestal extends 50 units downward from the hexagon's center with a diameter of 13.7 units."* Such descriptions capture modeling intent and scale information without exhaustively enumerating low-

---

[1]CadQuery exposes a subset of extrusion and revolve parameterizations. Our dataset is collected from Onshape and includes additional variants, which we represent through corresponding interfaces, e.g., a bidirectional revolve (*revolve_bidir*)

level parameters, encouraging the model to infer underlying parametric details from high-level design intent.

**Program Rewriting.** The initial CadQuery programs are obtained via procedural conversion. To diversify code representations, we prompt Claude-Sonnet-4.5 with relevant CadQuery documentation to rewrite them into more natural, reusable forms. Rewritten programs are validated via execution and filtered by visual similarity to original renderings, yielding about 80k additional samples. This encourages the model to learn more generalizable program structures.

**Training.** We perform supervised fine-tuning using a standard causal language modeling (CLM) objective. Given a textual description $x$ (either $x_a$ or $x_d$), the model is trained to autoregressively generate the corresponding CadQuery program $P$. Formally, the training objective is defined as

$$\mathcal{L}_{\text{SFT}} = -\mathbb{E}_{(x,P)} \frac{1}{T} \sum_{t=1}^{T} \log \pi_\theta(P_t \mid x, P_{<t}), \qquad (12)$$

where $T$ is the token sequence length of the program, and $\pi_\theta$ denotes the autoregressive LLM parameterized by $\theta$.

## 5.2. Reinforcement Learning Stage

We further optimize the model using reinforcement learning with a geometric reward and sequence-level policy optimization to improve generalization beyond supervised training.

**Group Sequence Policy Optimization.** We employ Group Sequence Policy Optimization (GSPO) (Zheng et al., 2025) in the reinforcement learning (RL) stage. Unlike token-level policy optimization methods such as GRPO (Shao et al., 2024), GSPO formulates the importance ratio from sequence likelihood and performs clipping, reward calculation, and optimization over entire sequences. This design enables stable training and efficient optimization by directly aligning policy updates with the sequence-level objectives:

$$\mathcal{L}_{\text{GSPO}}(\theta) = -\frac{1}{N} \sum_{i=1}^{N} \min \left( s_i(\theta) \widehat{A}_i, \, \text{CLIP}(s_i(\theta), \varepsilon) \widehat{A}_i \right),$$

$$(13)$$

where $N$ denotes the number of sampled responses $\{y_i\}$ for a given input prompt $x$. The importance ratio $s_i(\theta) = \left( \pi_\theta(y_i|x)/\pi_{\theta_{\text{old}}}(y_i|x) \right)^{1/|y_i|}$ is computed from the sequence-level likelihoods under the current and reference policies, normalized by the response length $|y_i|$. The clipping operator is defined as $\text{CLIP}(s_i(\theta), \varepsilon) = \text{clip}(s_i(\theta), 1-\varepsilon, 1+\varepsilon)$, where $\varepsilon$ denotes the clipping range of importance ratios. The group-normalized advantage estimator is given by:

$$\widehat{A}_i = \frac{r(x, y_i) - \text{mean}\left(\{r(x, y_i)\}_{i=1}^{N}\right)}{\text{std}\left(\{r(x, y_i)\}_{i=1}^{N}\right)}, \qquad (14)$$

where $r(x, y)$ denotes a scalar reward.

**Reward Design.** Similar to CAD-Coder (Guan et al., 2025), our reward function is based on the Chamfer Distance (CD) between the generated CAD model and the ground-truth model. For each sampled response $y_i$, we first extract the executable CAD program and execute it to obtain the resulting geometry. If execution succeeds, the resulting shape is uniformly sampled into a point cloud $X$, and the ground-truth CAD model is sampled into a reference point cloud $Y$. The Chamfer Distance between the two point clouds is computed as:

$$\text{CD}(X,Y) = \frac{1}{|X|} \sum_{x \in X} \min_{y \in Y} \|x - y\|_2^2 + \frac{1}{|Y|} \sum_{y \in Y} \min_{x \in X} \|x - y\|_2^2,$$

$$(15)$$

where $|X|$ and $|Y|$ denote the number of sampled points. Then, we convert the Chamfer Distance into a scalar geometric reward. Specifically, the reward assigns higher scores to geometrically closer shapes and linearly penalizes deviations beyond a tolerance threshold $t$:

$$r(x, y_i) = 1 - \frac{\min(\text{CD}(X,Y), t)}{t}, \qquad (16)$$

where $t = 0.2$ in our experiments. If the extracted program fails to execute, the reward is set to zero.

# 6. Experiments

## 6.1. Experimental Setups

**Datasets.** We conduct experiments on our FutureCAD dataset, which contains approximately 140k CAD entries. We split the dataset into 90% for training, 5% for validation, and 5% for testing. The dataset consists of two subsets: about 28k entries involve **advanced** features (i.e., *chamfer*, *fillet* and *shell*), while the remaining entries are **standard** CAD models without such advanced operations. For evaluation, we report results separately on the advanced and standard subsets to better characterize performance under different feature complexities. All evaluations are performed by providing the model with the detailed textual description as input. Additionally, we evaluate out-of-distribution generalization on the Fusion360 dataset (Willis et al., 2021), which contains only standard CAD models. Note that we do not evaluate on the DeepCAD dataset (Wu et al., 2021), as both our dataset and DeepCAD are collected from Onshape, and the standard subset of FutureCAD subsumes the DeepCAD data. More detailed information about our dataset is provided in Appendix A.

**Metric.** For text-to-CAD generation, we follow CADFusion (Wang et al., 2025a) and report the command-level F1 score. We further decompose it into **G-F1**, which evaluates geometry-constructing commands (*sketch*, *extrude*, and *revolve*), and **R-F1**, which evaluates refinement commands (*chamfer*, *fillet*, and *shell*). We also report the **Chamfer Distance (CD)** to assess geometric similarity between the generated and ground-truth shapes, and the **Invalidity Ratio**

| Methods | FutureCAD Dataset | | | | | | | | | Fusion 360 (Out-of-Distribution) | | | |
| --- | --- | --- | --- | --- | --- | --- | --- | --- | --- | --- | --- | --- | --- |
| | Advanced | | | | | Standard | | | | | | | |
| | G-F1↑ | R-F1↑ | Median CD↓ | Mean CD↓ | IR↓ | G-F1↑ | Median CD↓ | Mean CD↓ | IR↓ | G-F1↑ | Median CD↓ | Mean CD↓ | IR↓ |
| CAD-Translator | - | - | - | - | - | 79.08 | 94.23 | 152.60 | 3.57 | 76.82 | 159.98 | 211.72 | 3.86 |
| Text2CAD | - | - | - | - | - | 81.39 | 77.65 | 132.04 | 1.90 | 80.24 | 112.77 | 178.42 | 3.56 |
| Text-to-CadQuery | - | - | - | - | - | 87.04 | 109.83 | 173.69 | 8.38 | 84.36 | 188.88 | 232.56 | 11.70 |
| cadrille | - | - | - | - | - | 84.13 | 71.89 | 125.58 | 2.57 | 84.53 | 143.04 | 206.93 | 5.02 |
| CADFusion[†] | 79.02 | **91.12** | 54.14 | 97.09 | 50.60 | **91.91** | 42.61 | 91.86 | 10.76 | 89.32 | 41.48 | 102.06 | 12.15 |
| CAD-LLaMA[†] | 80.05 | 87.60 | 50.16 | 91.52 | 46.60 | 91.38 | 37.47 | 77.10 | 5.30 | 90.15 | 39.27 | 113.90 | 8.13 |
| *FutureCAD* (Ours) | **80.54** | 85.35 | **31.12** | **71.39** | **1.01** | 90.39 | **14.68** | **58.35** | **0.91** | **90.34** | **17.01** | **85.65** | **1.00** |

*Table 1.* Main results on text-to-CAD generation. † indicates that we extend the command space of the method to support advanced features and retrain on our dataset, while other methods are evaluated only on the standard subset due to lack of such support. Results are reported on our FutureCAD, as well as the out-of-distribution Fusion360 test set. *G*-F1 covers generation commands (*sketch*, *extrude*, *revolve*); *R*-F1 covers refinement commands (*fillet*, *chamfer*, *shell*). CD is multiplied by $10^3$ and all other metrics are multiplied by 100%.

**(IR)** to measure the fraction of generated CAD programs or resulting solids that are invalid (e.g., non-executable code, CAD-kernel failures, or non-manifold geometry). More details on metric computation are provided in Appendix C.

For *BRepGround* evaluation, we report **Recall@k**, **mAP**, and **F1**. Recall@k measures whether the ground-truth primitives are retrieved within the top-$k$ predictions, mAP summarizes ranking quality across queries, and F1 balances precision and recall for primitive selection.

**Implementation Details.** The backbone LLM for CAD program generation is Qwen2.5-7B-Instruct (Qwen et al., 2025). Supervised fine-tuning is performed for 3 epochs with a learning rate of $1 \times 10^{-5}$, followed by reinforcement learning (RL) with a learning rate of $1 \times 10^{-6}$. The RL procedure is built upon veRL (Sheng et al., 2025), where we sample 8 candidate programs per prompt during rollout for policy optimization. For *BRepGround*, we use a shared feature dimension of 1024 for both text and B-Rep representations. The fusion module has 8 layers with 8 attention heads and feed-forward dimension 4096. Optimization uses AdamW with learning rate $1 \times 10^{-4}$ and weight decay $5 \times 10^{-5}$, and the model is trained for 40 epochs.

**Baselines.** For the text-to-CAD task, we compare against CAD-Translator (Li et al., 2024) and Text2CAD (Khan et al., 2024b), as well as LLM-based approaches including Text-to-CadQuery (Xie & Ju, 2025), cadrille (Kolodiazhnyi et al., 2025), CADFusion (Wang et al., 2025a), and CAD-LLaMA (Li et al., 2025b). To enable comparison on the advanced subset, we extend CADFusion and CAD-LLaMA to support advanced feature execution by augmenting their command space with primitive selection: the model predicts a sequence of query points, and edge or face primitives are obtained by iteratively retrieving the nearest primitive of the required type in the B-Rep. The remaining baselines are evaluated only on the standard subset.

For the text-based B-Rep grounding task, we compare with three baselines. *LLM-Query* prompts an LLM to generate code that calls CadQuery native selector APIs for primitive selection. *CLIP-DE* is a dual-encoder trained with a CLIP-style contrastive loss and ranks primitives by similarity. *Late Fusion* encodes text and primitives separately and predicts per-primitive scores with a lightweight fusion head. More details of baselines are provided in Appendix D.

### 6.2. Main Results

**Quantitative Results.** The main results on the text-to-CAD task are reported in Table 1. On the advanced subset, our method achieves the lowest Chamfer Distance (CD), indicating higher-fidelity geometry, and attains a very low invalidity ratio of 1.01%, demonstrating a substantial improvement over baseline methods. On the standard subset, where CAD models are typically less complex, our method reduces CD while maintaining a low invalidity ratio of 0.91%, with the median CD improving from 37.47 to 14.68. Moreover, on the out-of-distribution Fusion360 test set, which also contains only standard CAD models, our method preserves a comparable level of performance to in-distribution evaluation, highlighting strong generalization. Although CADFusion and CAD-LLaMA achieve competitive command-type F1 scores, they remain significantly worse in CD and IR. This suggests that these methods could predict the command types correctly but struggle with accurate parameter prediction. In contrast, our approach better leverages the LLM's intrinsic reasoning and code-generation capability, enabling more precise parameterization and consequently higher geometric fidelity and validity.

**Qualitative Results.** Figure 4 presents qualitative comparisons on both the advanced and standard subsets of our FutureCAD dataset. On the Advanced subset, our method accurately captures fine-grained geometric details such as fillets and chamfers, whereas baseline methods often fail to generate valid models or produce outputs lacking these refinement details. *BRepGround* contributes to this by precisely grounding textual queries to target B-Rep primitives. On the Standard subset, our method consistently generates CAD models that closely match the textual specifications. These qualitative results corroborate our quantitative findings, showing that FutureCAD achieves superior geometric fidelity across varying model complexities.

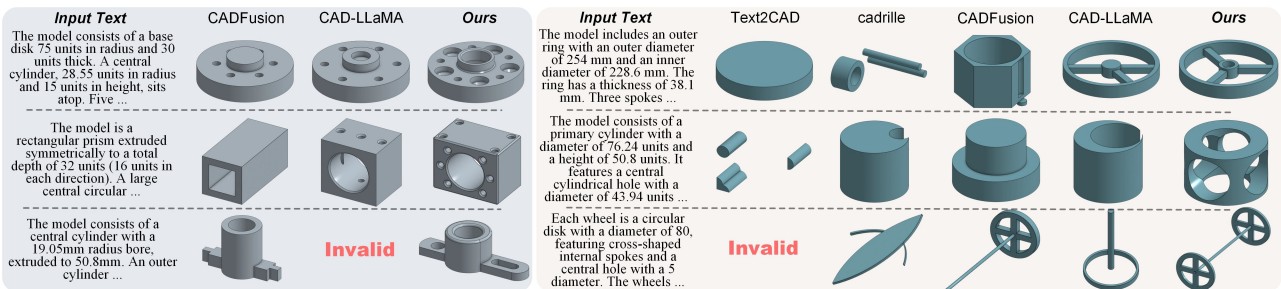

*Figure 4.* Qualitative comparison of text-to-CAD generation. ***Left:*** Results on the advanced subset, which includes models with advanced operations. ***Right:*** Results on the standard subset. Our method produces more accurate CAD models aligned with input descriptions.

| Methods | Recall@3↑ | Recall@5↑ | Recall@10↑ | mAP↑ | F1↑ |
|---|---|---|---|---|---|
| LLM-Query | - | - | - | - | 21.51 |
| CLIP-DE | 51.81 | 62.57 | 75.01 | 57.53 | 31.05 |
| Late Fusion | 51.43 | 63.55 | 76.72 | 58.18 | 43.32 |
| Ours | **56.39** | **65.86** | **78.47** | **63.07** | **50.23** |

*Table 2.* Results on text-based B-Rep grounding task. LLM-Query prompts an LLM to generate code using CadQuery native selector APIs. CLIP-DE trains a dual encoder with CLIP-style contrastive loss. Late Fusion concatenates pooled text features with primitive embeddings for classification. LLM-Query produces binary outputs and is excluded from ranking metrics. All metrics are multiplied by 100%

## 6.3. BRepGround Evaluation

Table 2 reports results on the text-based B-Rep grounding task. Our method achieves 63.07% mAP and 50.23% F1, outperforming all baselines by a significant margin. The LLM-based approach achieves only 21.51% F1, as limited selector expressiveness and the inability of LLMs to perceive B-Rep geometry hinder precise translation from natural language to primitive selection. Learning-based methods improve over LLM-Query, yet both CLIP-DE and Late Fusion show a gap between ranking and classification performance. This discrepancy suggests that while these methods capture coarse semantic relevance, they lack the discriminative capacity to precisely delineate target primitives from distractors. The grounding task requires fine-grained alignment between language and geometry, as queries often combine constraints on shape, position, and topology. Compressing queries into a single vector, as done by CLIP-DE and Late Fusion, loses compositional information, while scoring primitives in isolation ignores structural context from neighboring faces and edges. Our method preserves the multi-faceted nature of queries through cross-attention and captures inter-primitive dependencies via self-attention, as validated by consistent gains across all metrics.

## 6.4. Ablation Studies

We conduct ablation studies on the advanced subset to validate our design choices. Table 3 compares three configurations: supervised fine-tuning alone (SFT *only*), replacing *BRepGround* with point-based nearest neighbor selection

| Methods | G-F1↑ | R-F1↑ | Median CD↓ | Mean CD↓ | IR(%)↓ |
|---|---|---|---|---|---|
| SFT *only* | 79.82 | **91.44** | 44.00 | 85.15 | 42.67 |
| *w/o* BRepGround | 80.36 | 82.22 | 38.50 | 76.01 | 8.29 |
| Ours | **80.54** | 85.35 | **31.12** | **71.39** | **1.01** |

*Table 3.* Ablation study for text-to-CAD generation on the advanced subset. *w/o* BRepGround replaces text-based primitive selection with point-based nearest neighbor retrieval.

(*w/o* BRepGround), and our full method. Supervised fine-tuning alone exhibits a high invalid rate despite reasonable sequence accuracy, as the model lacks feedback on operation executability. Incorporating reinforcement learning substantially reduces invalid outputs by penalizing failed executions. However, point-based selection remains error-prone due to its limited robustness to geometric ambiguity, such as cases where multiple primitives cluster near the query point. Our *BRepGround* resolves such ambiguities by leveraging text semantics for primitive selection, achieving the lowest invalid rate and best geometric fidelity. Despite these advances, our method still has limitations; we discuss failure cases in Appendix E.

## 7. Conclusion

We present FutureCAD, a text-to-CAD framework that unifies LLM-based program generation with explicit B-Rep primitive grounding. By introducing *BRepGround*, a transformer-based module that grounds textual queries to geometric primitives, our method enables the generation of realistic CAD models involving advanced features such as *fillet*, *chamfer*, and *shell*. We further contribute a dataset of over 140k real-world CAD models, annotated with both geometric descriptions for text-to-CAD generation and textual queries for text-based B-Rep grounding. For the LLM, we adopt a two-stage training strategy combining supervised fine-tuning with reinforcement learning. Experiments show that FutureCAD achieves state-of-the-art performance on both in-distribution and out-of-distribution benchmarks, substantially improving geometric fidelity while maintaining a low invalid rate. The framework integrates seamlessly with modern feature-based CAD systems, facilitating practical deployment in real-world design workflows.

## Acknowledgement

The computations in this research were performed using the CFFF platform of Fudan University.

## Impact Statement

This paper advances machine learning for text-to-CAD generation, aiming to reduce the effort required to create parametric CAD models from natural-language descriptions. The potential benefits include faster prototyping and improved accessibility of CAD tools for non-expert users. Potential risks include misuse to facilitate the design of harmful or regulated objects, and the generation of incorrect designs that could be unsafe if directly manufactured. We encourage human-in-the-loop review and domain-specific safety checks before downstream use.

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

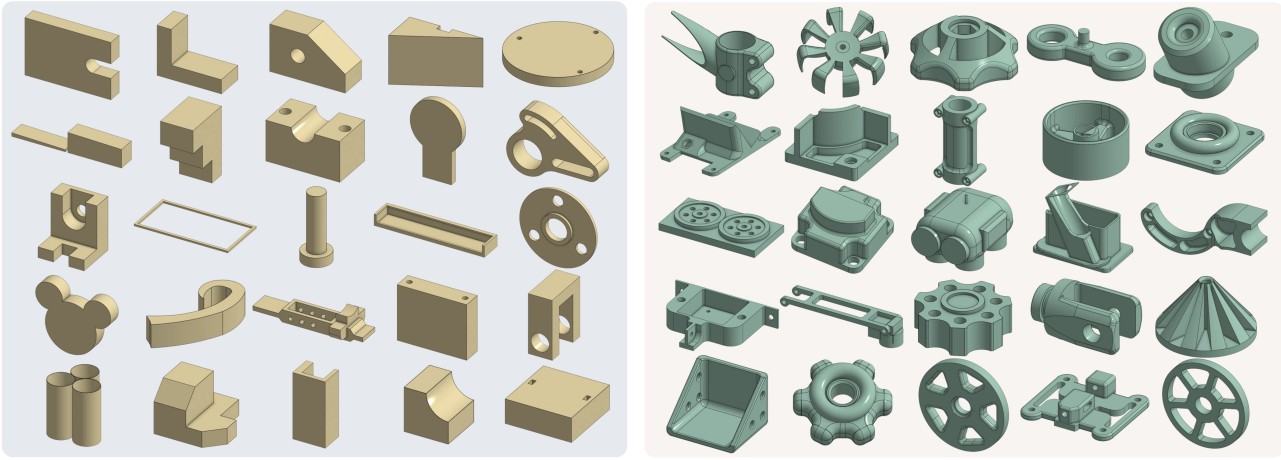

DeepCAD Dataset                    FutureCAD Dataset (Ours)

*Figure 5.* **Comparison of DeepCAD and FutureCAD datasets.** DeepCAD (left) contains simple geometric primitives, while FutureCAD (right) features complex real-world industrial parts with advanced operations such as *fillet*, *chamfer*, and *shell*.

## A. Dataset Details

Our dataset is constructed by parsing CAD models from the ABC dataset (Koch et al., 2018) using FeatureScript files and the Onshape developer API. The dataset supports the following feature types: *sketch*, *extrude*, *revolve*, *fillet*, *chamfer*, and *shell*. Since the DeepCAD dataset is also derived from ABC and only supports *sketch* and *extrude* operations, our dataset subsumes the DeepCAD dataset. The ABC dataset consists of real-world CAD models created by human designers and contains substantial redundancy due to model reuse. We therefore perform deduplication based on rendered image similarity to ensure diversity.

We note that a concurrent work, WHUCAD (Fan et al., 2025), also proposes a dataset with advanced features. However, as reported in their paper, the dataset contains 146k models in total, among which approximately 4k are manually constructed, while the remaining majority are procedurally augmented from DeepCAD, potentially limiting how well the dataset captures real-world industrial CAD design practices. Furthermore, WHUCAD provides CAD models only as parametric sequence matrices and relies on a proprietary cloud-based interface with CATIA as the kernel for rendering and export, without offering processing scripts compatible with open-source CAD kernels (e.g., OpenCASCADE), limiting its reusability. Another work, Seek-CAD (Li et al., 2025c), supports execution on open-source CAD kernels but remains constrained by its *CapType* design paradigm, which is inherently unable to reference primitives generated by Boolean operations during the modeling process. This fundamentally limits its applicability to real-world CAD scenarios.

In contrast, our dataset contains CAD models that are all manually created by human designers rather than procedurally augmented, spanning diverse levels of complexity with a balanced distribution. We additionally provide multiple representation formats, including JSON and CadQuery program, and support parsing with open-source CAD kernels, enabling convenient use in future work.

To illustrate the differences between our dataset and existing datasets, we provide both quantitative and qualitative comparisons. As shown in Figure 6, our FutureCAD dataset exhibits a more balanced distribution across varying levels of design complexity compared to DeepCAD, with a substantially higher proportion of models containing longer feature sequences. Figure 5 further shows that DeepCAD models are predominantly simple geometric primitives, whereas FutureCAD encompasses diverse industrial parts with intricate details. These characteristics make our dataset a more challenging and practical benchmark for AI-driven CAD generation research.

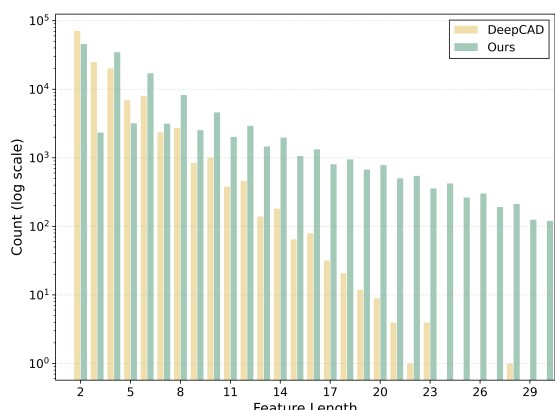

*Figure 6.* Feature length distribution comparison between DeepCAD and our FutureCAD dataset.

## B. Prompt Templates for Data Generation

For reproducibility, we provide the prompt templates used in the LLM-assisted data generation pipeline. These prompts are used for generating textual descriptions of CAD models and text queries for B-Rep primitive grounding.

**Textual description generation.** Given an image and the CAD code that generates it, the LLM is prompted as follows:

```
You are given an image and the CAD code that generates it.
Your task is to write two levels of geometric descriptions:

- Overall Description (max 60 words): Provide a high-level summary of the model's
↪  geometry.
- Detail Description (max 120 words): Provide a precise description of the model's
↪  geometry, including dimensions, proportions, and significant geometric features as
↪  reflected in the code. If the code contains many parameters, focus on key dimensions
↪  such as overall width, height, depth, and counts of repeating features.

Notes:
- Do NOT describe colors, materials, lighting, rendering effects, or coordinate
↪  information.
- Base your geometric descriptions primarily on the visual information from the image. The
↪  code should be used as a reference for dimensions, quantities, or overall structural
↪  hints.

The CAD Code:
{cad_code}

Your answer must be strictly in the following JSON format:
```json
{{
  "overall_description": "your overall description here",
  "detail_description": "your detailed description here"
}}
```
```

**Primitive query generation.** For features that require references to B-Rep primitives, we render the original CAD model and another view in which the selected edges or faces are emphasized. The LLM is prompted to describe only the selected geometry:

```
You are a CAD B-Rep grounding assistant.

Your task is to describe ONLY the selected geometry (edges and/or faces) in a 3D CAD model
↪  using purely geometric language. The description will be used for a grounding task:
↪  given a B-Rep and text, the model predicts the selected edge/face labels.

Inputs:
1. Image 1 - Original CAD model
2. Image 2 - Same model with the selected edges/faces emphasized in red. The red
↪  highlighting is only a visual cue to identify the selection; do NOT mention color or
↪  highlighting.

Additionally provided:
- Meta information (auxiliary, for reasoning only):
  - Number of selected edges: {edge_count}
  - Number of selected faces: {face_count}

The meta information is not required to be explicitly stated in the output. It is only
↪  provided to help infer which geometry is selected.

Task:
Describe the selected edges and/or faces ONLY.
Do NOT describe attributes irrelevant or harmful to the grounding task, such as color,
↪  material, or texture.

Description guidelines:
```

```
Use natural, location-focused language. The goal is to provide sufficient semantic cues to
↪  uniquely localize the selected edges/faces within the entire model. Descriptions may
↪  include:
- where the selected geometry is located on the model as a whole
- how it is positioned relative to the main body or major features
- whether it lies on an outer boundary, inner region, corner, extremity, or transition
↪  area
- how it relates spatially to nearby prominent structures such as holes, protrusions, or
↪  recesses

Use clear, human-like geometric descriptions that help point to the selected faces/edges.
Avoid abstract taxonomies or rigid category listings; focus on where it is and how to find
↪  it.

Output format:
Respond in JSON with exactly one key:
```json
{{
  "descriptions": [
    "Geometric description from one viewpoint",
    "Geometric description from a different viewpoint",
    "Geometric description from another viewpoint"
  ]
}}
```

Additional requirements:
- Avoid redundant wording.
- Each description must be no longer than 100 words.
- Use compact, information-dense language.
```

**Examples of generated grounding queries.** Examples of the resulting text queries used to train *BRepGround* include:

1. The eight hole rim edges positioned on the upper surface of the part, each marking the circular opening of a cylindrical void that passes completely through the block's thickness.
2. The top horizontal edge of the vertical post, where the upper face meets the back face at the highest point of the structure.
3. The complete outer boundary surfaces of the cube excluding the top face: comprising the four vertical side walls and the bottom base that collectively form five of the six faces of the rectangular solid.

## C. Metric Details

In this section, we provide additional details on the metric computation for the text-to-CAD task. For the computation of command-level F1 score, we follow the open-source implementation provided by (Wang et al., 2025a). Regarding Chamfer Distance, we note that different implementation choices can lead to substantially different numerical results. To ensure fair and consistent comparison across all experiments, we adopt a unified Chamfer Distance computation protocol. Specifically, for each CAD model, we uniformly sample 2048 points from the surface. The sampled points are then translated to be centered at the origin by subtracting the centroid and scaled by the maximum Euclidean distance from the origin. This normalization ensures that all points lie within a unit sphere, effectively mapping the geometry to the range $[-1, 1]$. The Chamfer Distance is then computed on the normalized point clouds. Finally, the Invalidity Ratio (IR) is determined based on whether the generated CAD model is geometrically and topologically valid. We utilize the APIs provided by OpenCASCADE to check the validity of the resulting shape. A model is considered invalid if the generated program fails to execute or if the resulting shape does not pass the validity checks of the CAD kernel. Due to the limited robustness of OpenCASCADE compared to commercial kernels, we adopt a lenient evaluation protocol for advanced features: a feature is considered invalid only when all its referenced primitives fail to execute, while partial execution failures are tolerated.

To examine the impact of this lenient protocol, we additionally re-evaluate all 7,168 advanced test samples under a strict validity criterion, where any primitive execution failure in an advanced operation is counted as invalid. Under this strict protocol, only 2 additional samples are marked as invalid compared with the lenient protocol. Consequently, the invalidity ratio on the advanced subset remains 1.01% after rounding, indicating that the lenient protocol does not materially affect

our conclusions. We retain the lenient protocol because some advanced operations, such as fillet and chamfer on complex geometry, may succeed in commercial CAD kernels (e.g., Parasolid kernel) but fail in OpenCASCADE due to kernel robustness limitations.

## D. Baseline Implementation Details

### D.1. Text-to-CAD Baselines

For CADFusion and CAD-LLaMA, we extend their original command sets to align with our dataset and train the models following the procedures described in their respective papers. For the remaining baselines, evaluation is restricted to the intersection of our *Standard* subset and the DeepCAD dataset, which contains approximately 80k samples, since these methods support only *sketch* and *extrude* operations. For CAD-Translator and Text2CAD, we follow their CAD sequence representation. For Text-to-CadQuery and cadrille, we directly adopt their open-source data formats.

### D.2. Text-based B-Rep Grounding Baselines

**LLM-Query.** For *LLM-Query*, we prompt Claude-Sonnet-4.5 (Anthropic, 2025) with multi-view renderings of the input B-Rep, the textual query, and the CadQuery selector API documentation. The model is instructed to generate CadQuery selector code, from which the predicted primitives are extracted via post-processing and compared with the ground truth.

**CLIP-DE.** For *CLIP-DE*, we adopts a dual-encoder architecture that embeds the textual query and each B-Rep primitive into a shared representation space, and performs grounding via similarity matching. Given a B-Rep, the model encodes all face and edge primitives into a set of UV-token embeddings $\{u_i\}_{i=1}^{L}$, while the textual query is encoded into a single global text embedding $t$ obtained by mean pooling over token-level representations. All embeddings are $\ell_2$-normalized. The similarity between a primitive $u_i$ and the textual query is computed as:

$$s_i = \frac{u_i^\top t}{\tau}, \tag{17}$$

where $\tau$ denotes a temperature parameter.

Training is supervised using an in-sample bidirectional CLIP-style contrastive objective. Primitives with ground-truth label 1 are treated as positive samples, while those with label 0 are treated as negatives. The text-to-primitive loss encourages the textual query to be more similar to all positive primitives than to negative ones:

$$\mathcal{L}_{\text{text}\to\text{prim}} = -\frac{1}{|\mathcal{P}|} \sum_{i \in \mathcal{P}} \log \frac{\exp(s_i)}{\sum_{j \in \mathcal{P} \cup \mathcal{N}} \exp(s_j)}, \tag{18}$$

where $\mathcal{P}$ and $\mathcal{N}$ denote the sets of positive and negative primitives, respectively.

The primitive-to-text loss further enforces each positive primitive to be more similar to the query than all negative primitives:

$$\mathcal{L}_{\text{prim}\to\text{text}} = -\frac{1}{|\mathcal{P}|} \sum_{i \in \mathcal{P}} \log \frac{\exp(s_i)}{\exp(s_i) + \sum_{j \in \mathcal{N}} \exp(s_j)}. \tag{19}$$

The final training objective is defined as

$$\mathcal{L}_{\text{CLIP-DE}} = \frac{1}{2} \left( \mathcal{L}_{\text{text}\to\text{prim}} + \mathcal{L}_{\text{prim}\to\text{text}} \right). \tag{20}$$

At inference time, primitives are ranked according to their similarity scores $s_i$.

**Late Fusion.** *Late Fusion* replaces the fusion module in *BRepGround* with a late-fusion design based on global text pooling and feature concatenation. Given UV-token embeddings of B-Rep primitives $\{u_i\}_{i=1}^{L}$ and a global text embedding $t$ obtained by mean pooling. The $t$ is then concatenated with each primitive embedding and passed through a multilayer perceptron to obtain fused features:

$$h_i = \text{MLP}\big([u_i; t]\big), \tag{21}$$

where $[\,\cdot\,;\,\cdot\,]$ denotes feature concatenation. Finally, a matching head maps each fused feature to a scalar grounding score:

$$\hat{s}_i = g(h_i). \tag{22}$$

# E. Failure Cases

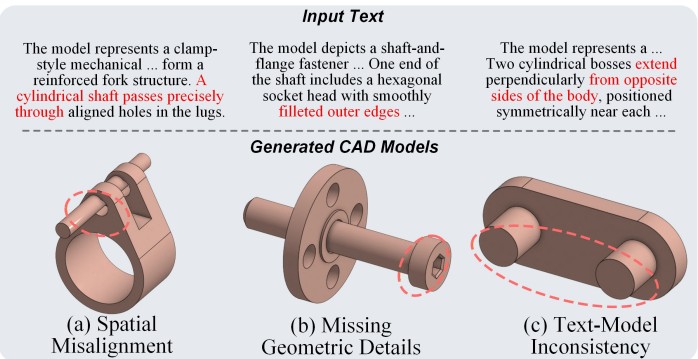

*Figure 7.* Typical failure cases of our method. (a) Spatial Misalignment: incorrect positioning of geometric primitives. (b) Missing Geometric Details: omission of specified refinement features such as fillets. (c) Text-Model Inconsistency: generated geometry contradicts the spatial arrangement described in the input text.

While our method achieves strong performance overall, we observe several representative failure cases, as shown in Figure 7, which expose current limitations of LLM-driven CAD generation and text-based B-Rep grounding.

Figure 7(a) illustrates a spatial misalignment failure, where geometric primitives are generated with incorrect relative positioning. This behavior can be attributed to the limited capability of large language models in precise three-dimensional spatial reasoning. Although LLMs are effective at generating syntactically valid programs, reasoning about exact spatial relationships and alignment constraints remains challenging. Figure 7(b) presents a case where refinement features such as fillets are omitted despite being specified in the input description. A potential contributing factor is the class imbalance inherent in the text-based B-Rep grounding task, where refinement-related primitives constitute a small fraction of the overall B-Rep. This imbalance biases the model toward negative predictions, reducing sensitivity to fine-grained refinement features. Figure 7(c) shows a text-model inconsistency, in which the generated geometry contradicts the spatial arrangement described in the input text. This suggests that the model does not always successfully align fine-grained linguistic descriptions with their corresponding code-level representations, particularly when subtle spatial semantics must be translated into executable CAD operations.

With continued advances in methods for enhancing spatial reasoning in large language models, together with the availability of richer and more diverse training data, we expect these limitations to be progressively alleviated.

