# OpenReview forum: "Towards High-Fidelity CAD Generation via LLM-Driven Program Generation and Text-Based B-Rep Primitive Grounding"
_ICML.cc/2026/Conference — ICML 2026 spotlight_

### Official Review · Reviewer_g5Ps · 2026-03-09

**Soundness:** 2
**Presentation:** 3
**Significance:** 3
**Originality:** 3
**Overall Recommendation:** 4
**Confidence:** 4

**Summary:**

In FutureCAD, the authors proposed an LLM framework with a novel module called **BRepGround** to generate high-fidelity CAD models from text descriptions. LLMs generate CADQuery codes from text prompts with advanced operations like fillet and chamfer. However, these operations require selecting specific faces/edges on the 3D shape, which BRepGround handles by matching natural language queries to geometric primitives using cross-attention between text and shape encodings. They also contribute a new dataset of 140k real-world CAD models, more complex than the DeepCAD dataset.

**Compliance With Llm Reviewing Policy:**

Affirmed.

**Final Justification:**

My primary concerns were regarding the experiments which have been addressed. Therefore I am raising the score.

**Key Questions For Authors:**

I have several issues.

- The authors used Distance-based metrics (CD, F1) in Table 1. However, these are not reliable metrics at all in the text-to-CAD as well as in the text-to-3D domain. Consider "a simple cylindrical shape." If model X generates a cylinder but rotated $90\degree$ along x axis, it will have a higher CD compared to model Y who has generated a rectangular shape with the same orientation. For evaluation, human and VLM (GPT, Claude) are the most reliable choices.

- Continuing on the issue with distance metrics, during RL, authors use CD for reward computation. Consider the following example.

Scenario: "A flat circular disk with 6 evenly spaced small holes around the perimeter."

During RL rollout, the model generates two candidate programs:

Sample A: Generates the correct flat disk with 6 holes, but the disk is thicker than ground truth, giving a large CD because the extra thickness shifts thousands of surface points away from the reference.

Sample B: Generates a plain flat disk with no holes but with dimensions exactly matching the ground truth thickness and diameter, giving a small CD because the outer surface aligns nearly perfectly, and the 6 small holes contribute negligible surface area to the point cloud.

A better choice will be visual (large VLM) feedback.

- How many predictions per prompt were generated during Inference (For FutureCAD, CAD-LLAMA, and CADFusion)? This is a critical missing detail for reproducibility and fair comparison on the Invalidity Ratio. CADQuery-based models are notorious for a large number of invalid models.

Overall, the experimental section requires revision before the reported results can be considered reliable. I am currently at a borderline position and look forward to the authors' response during the discussion phase to clarify these concerns.

**Limitations:**

Yes, the authors have provided sufficient limitations and failure cases in the supplementary material.

**Strengths And Weaknesses:**

## Strengths

- The primary technical novelty is BRepGround, which identifies specific faces/edges for advanced CAD operations via text-geometry grounding.

- The FutureCAD dataset will be very useful for the community only if it's released.

- The paper is mostly well written with good figures.

## Major Weaknesses

Please check questions sections for more issues.

- The assumption that "they fail to support advanced operations such as fillet and chamfer" (line 42-46, 2nd paragraph) is not correct. CADParser [1] demonstrates that fillet and chamfer operations can be supported within a parametric modeling framework, alongside a corresponding dataset.

- The individual components (UV-Net for B-Rep encoding, BERT for text encoding, transformer cross-attention for fusion, GSPO for RL) are all existing techniques. The originality lies in their combination and application, which is well-motivated but not architecturally groundbreaking.


## Minor Weaknesses

For better readability. No effect on final score.

- The motivation for BRepGround is insufficiently established in the Introduction. In the second paragraph, it should be clear to new readers. You might say that "However, they fail to support
advanced operations such as fillet and chamfer, which rely
on B-Rep primitives.." Here you should explicitly say faces or edges (Non-CAD readers might not understand BRep primitive part).

- The double bracket notation ⟦P⟧ in Equation 4 is used without any explicit definition or citation. While readers familiar with programming language theory may recognize it as denotational semantics, a brief clarification would improve accessibility for the broader ML audience at ICML.


[1]  CADParser: A Learning Approach of Sequence Modeling for B-Rep CAD (https://www.ijcai.org/proceedings/2023/200)

---

> ### Author Rebuttal · Authors · 2026-03-30
>
> We sincerely thank the reviewer for the constructive and insightful comments. Below, we provide point-by-point responses to the concerns raised.
>
> ### **(1) Response to CADParser Concern**
>
> We thank the reviewer for pointing out CADParser. However, CADParser addresses a **fundamentally different task** and does not invalidate our claim.
>
> CADParser is a **reconstruction** method: given a *complete, finalized* B-Rep, it infers the construction sequence. FutureCAD is a **forward generation** method: given a text description, it must resolve B-Rep primitive references *on-the-fly* during sequential execution. The key differences are:
>
> | Dimension | CADParser | FutureCAD |
> |---|---|---|
> | **Task** | B-Rep → Sequence (reconstruction) | Text → CAD (generation) |
> | **B-Rep Availability** | Full B-Rep given as input -- **unavailable in generation**, where the target shape does not yet exist | Only **transient B-Rep** available at each step (matching real-world feature modeling) |
> | **Primitive Selection** | Heuristic 3D point coordinate (suitable only when target B-Rep is known) | Natural language queries grounded to transient B-Rep primitives via BRepGround |
> | **Generalization** | Requires complete target geometry a priori | Language-driven; generalizes to unseen shapes and descriptions |
>
> In summary, **CADParser relies on a strong prior (the complete target B-Rep) and does not address the core challenge of parametric modeling in a generative setting**-- the insights its framework offers are of very limited applicability to generation. Empirically, replacing BRepGround with heuristic nearest-neighbor selection increases invalidity (8.29% vs. 1.01%) and reduces geometric fidelity (Table 3), while our method achieves precise grounding and better generalization.
>
> ### **(2) Response to Architectural Novelty Concern**
>
> We would like to clarify the sources of novelty in FutureCAD, which extend beyond individual components.
>
> **The problem formulation and solution are novel.** We are the first to formally define the feature-based parametric modeling problem for AI-driven CAD generation and to introduce *text-based B-Rep primitive grounding* as a principled solution. Advanced operations (e.g., fillet, chamfer) require resolving references in a partially constructed B-Rep during execution, a challenge not addressed in prior work. The key innovation of our framework lies in its natural coupling of LLM-based code generation with on-the-fly semantic grounding from textual descriptions during CAD kernel execution, thereby providing a systematic solution to a critical bottleneck in the field.
>
> **The architecture of BRepGround is task-driven, not arbitrarily assembled.** UV-Net captures the continuous parametric geometry of B-Rep faces and edges; BERT encodes the compositional semantics of natural language queries; and cross-attention enables fine-grained token-level alignment between the two modalities. This design is validated empirically -- BRepGround substantially outperforms both CLIP-DE and Late Fusion, confirming that the architectural choices are non-trivial and well-justified.
>
> ### **(3) Response to CD Metric Reliability and RL Reward Design**
>
> **On evaluation metrics.** To provide a more comprehensive evaluation, we conducted an additional VLM-based assessment on LLM-based methods, following the protocol of CADFusion, scoring generated models on a 10-point scale. Results on our dataset are as follows:
>
> | Method | Text-to-CadQuery | cadrille | CADFusion | CAD-LLaMA | **FutureCAD (Ours)** |
> |---|---|---|---|---|---|
> | VLM Score | 6.10 | 6.43 | 7.57 | 7.63 | **8.04** |
>
> FutureCAD achieves the highest VLM score, consistent with our CD-based results, confirming robustness across evaluation protocols.
>
> **On CD as RL reward.** We appreciate the reviewer's example, which illustrates a valid theoretical concern. Our empirical results suggest that CD reward, while imperfect in theory, provides sufficient learning signal in practice: FutureCAD achieves the highest VLM score, improved geometric fidelity, and a substantially lower invalidity ratio, indicating that the model has learned to generate geometrically faithful and valid outputs. We acknowledge that the reviewer's disk-with-holes example represents a valid theoretical concern, and we agree that exploring VLM-based reward signals is a promising direction for future work.
>
> ### **(4) Response to Inference Sampling Details**
>
> Here we clarify the inference setup for all methods.
>
> For LLM-based methods, evaluation is conducted on a single A100 GPU using vLLM. We generate exactly **one** response per test entry for all methods (with temperature=0.7, top_p=0.9, and max_tokens=4096). For other baselines, we use the greedy decoding, and produce **one** response per sample. All metrics (including IR) are computed under a consistent protocol for fair comparison.
>
>
> We hope these results help address the reviewer's concerns.

---

> > ### Author Rebuttal · Reviewer_g5Ps · 2026-04-03
> >
> > My main concerns regarding the experiments have been resolved. I am raising the score.
> > Authors are reminded to open source the dataset for community benefits.

---

### Official Review · Reviewer_VGxa · 2026-03-11

**Soundness:** 3
**Presentation:** 3
**Significance:** 3
**Originality:** 3
**Overall Recommendation:** 4
**Confidence:** 3

**Summary:**

This paper presents FutureCAD, a text-to-CAD framework that bridges LLM-driven program generation with text-based B-Rep primitive grounding. The key observation is that advanced CAD operations like fillet, chamfer, and shell require selecting specific geometric primitives (faces, edges) from the intermediate B-Rep, something that purely parametric sequence models cannot express. FutureCAD addresses this by having the LLM generate CadQuery programs that embed natural-language queries (e.g., filletQ(cq.Query("the boundary loops where the flanged body and the recessed central..."), 3.54)) wherever primitive references are needed. A dedicated transformer model called BRepGround then resolves these queries against the transient B-Rep state at execution time, selecting the target faces or edges. The authors construct a dataset of approximately 140k real-world CAD models from the ABC dataset and Onshape, with about 28k involving advanced features, annotated with geometric descriptions generated by Claude-4.5 and converted into CadQuery programs. The LLM (Qwen2.5-7B-Instruct) is trained via supervised fine-tuning followed by reinforcement learning using GSPO with a Chamfer Distance-based reward. Experiments on in-distribution and out-of-distribution (Fusion360) benchmarks show strong geometric fidelity and low invalidity rates compared to six text-to-CAD baselines. BRepGround also outperforms three grounding baselines on the primitive selection task.

**Compliance With Llm Reviewing Policy:**

Affirmed.

**Key Questions For Authors:**

1. Can you report IR under a strict validity criterion (any partial execution failure counts as invalid) alongside the current lenient protocol? Even as a supplementary table, this would help calibrate the 1.01% number.

2. The R-F1 drop from SFT only (91.44) to the full method (85.35) deserves explanation. Is RL finding alternative but valid command decompositions that differ from the ground truth, or is it genuinely degrading refinement command accuracy? An analysis of whether R-F1 errors correlate with CD improvements would clarify this.

3. How were the evaluation text descriptions obtained? If they are all Claude-4.5-generated from ground-truth programs, have you tested on any human-authored prompts, even informally? Performance on shorter or less precise descriptions would strengthen the practical relevance claims.

4. Could you evaluate a stronger advanced-feature baseline - for instance, giving CADFusion or CAD-LLaMA access to a trained grounding module (e.g., CLIP-DE or even BRepGround) instead of nearest-primitive retrieval? This would better isolate how much of the improvement comes from program generation versus grounding.

5. Can you report end-to-end runtime including BRepGround inference and CAD kernel execution? For interactive design applications, latency matters and is currently absent from the evaluation.

6. The RL reward is based purely on Chamfer Distance. Have you experimented with or considered topology-aware rewards (e.g., penalizing missing features or incorrect face counts)? CD alone does not distinguish between topologically correct and incorrect models when they happen to be geometrically close.

**Limitations:**

The paper does not include a dedicated limitations section, though the failure case analysis in Appendix D identifies three categories of errors: spatial misalignment, missing geometric details (e.g., omitted fillets despite being specified), and text-model inconsistency. These are useful. However, several evaluation-level concerns go unacknowledged. The lenient validity protocol for advanced features could meaningfully affect the reported IR numbers, but this is described only in the appendix without discussion of its impact. The exclusive reliance on LLM-generated descriptions for both training and evaluation is not flagged as a limitation, nor is the potential circularity of training on Claude-4.5 outputs and then evaluating on similar Claude-4.5-generated prompts. The nearest-primitive heuristic used to extend baselines to advanced features is acknowledged as limited but no stronger alternative is attempted. Runtime analysis is absent despite the multi-stage pipeline (LLM generation, BRepGround inference, kernel execution) being relevant for practical deployment.

**Strengths And Weaknesses:**

**Strengths:**

I think the core contribution here is well-identified and genuinely important. Existing text-to-CAD methods generate either parametric sequences (which cannot express fillet/chamfer/shell because these require B-Rep primitive references) or direct B-Rep (which sacrifices editability). FutureCAD's approach of having the LLM emit text queries that get resolved at execution time against the evolving B-Rep is a clean solution to this problem, and it mirrors how modern CAD kernels actually work - features are applied to selected geometry, not to abstract indices. The CadQuery extension with filletQ/chamferQ/shellQ that accepts natural-language selectors is an elegant design choice that keeps the LLM in its comfort zone (generating text) while delegating geometric reasoning to a specialized module.

BRepGround's architecture is well-motivated for the task. UV-Net for face and edge geometry, a GNN over the face-adjacency graph for structural context, BERT for the textual query, and cross-attention for fusion - each component addresses a specific aspect of the grounding problem. The dedicated evaluation of BRepGround against three baselines (Table 2) is useful: the LLM-Query baseline (prompting Claude-Sonnet-4.5 with multi-view renderings) achieves only 21.51% F1, confirming that current VLMs cannot reliably map text to B-Rep primitives. The learned approaches (CLIP-DE, Late Fusion) do better but BRepGround's cross-attention fusion and graph-based context give it a clear edge, particularly on F1 (50.23 vs 43.32 for Late Fusion). The ablation in Table 3 gives an honest picture: RL is critical for validity (IR drops from 42.67% to 1.01%), and replacing BRepGround with point-based nearest-neighbor selection still leaves IR at 8.29%. The Fusion360 out-of-distribution results are encouraging, with comparable CD and IR to the in-distribution standard subset.

**Weaknesses:**

My main concern is the evaluation protocol for validity on advanced features. Appendix B explains that a feature is considered invalid only when all its referenced primitives fail to execute, with partial execution failures tolerated. This means a fillet that successfully rounds 3 out of 5 specified edges counts as valid. The reported IR of 1.01% on the advanced subset is therefore computed under lenient conditions, and it would be informative to see what happens under a strict criterion where any partial failure counts as invalid. The paper does not report this, making it hard to calibrate how well the system actually handles the full specification of complex features.

The baselines for the advanced subset also raise a fairness question. CADFusion and CAD-LLaMA were not designed for advanced features and are extended via a nearest-primitive heuristic - the model predicts query points, then retrieves the nearest primitive of the required type from the B-Rep. This is a simple geometric proxy that lacks any semantic understanding of what "the recessed central face" refers to. Comparing this to FutureCAD's dedicated BRepGround module, which is specifically trained for text-conditioned primitive selection with cross-attention over the full B-Rep, is not a fair fight. A stronger baseline would be to give the extended methods access to a retrained dual-encoder (like CLIP-DE) or even FutureCAD's own BRepGround for primitive selection, isolating the contribution of the program generation component from the grounding component.

The text descriptions used for both training and evaluation are generated by Claude-4.5 from ground-truth CAD programs and rendered images. No human-authored prompts are included. This is a real question mark for practical applicability: the descriptions in the dataset are detailed and geometrically precise ("The hexagonal top has a span of approximately 40 units in width and 23.09 units in height, extruded upwards by 15 units"), which is not how most users would describe a part they want. Whether FutureCAD degrades gracefully on shorter, vaguer, or less geometrically specific prompts is unknown.

One more observation from the ablation: R-F1 actually drops from 91.44 (SFT only) to 85.35 (full method with RL), even as CD and IR substantially improve. The paper does not discuss this. It suggests RL pushes the model toward geometrically closer outputs that may use different command decompositions than the ground truth, which could be benign. But it could also mean RL is trading off command-level accuracy for geometric reward, and understanding which is happening matters for assessing the RL stage.

---

> ### Author Rebuttal · Authors · 2026-03-30
>
> We thank the reviewer for the constructive feedback and helpful suggestions. We address each point below.
>
> ### **(1) Strict Validity Criterion**
>
> We re-evaluated all **7,168 test samples** under the **strict criterion (any primitive execution failure counts as invalid)**. The number of additionally invalidated samples is **only 2**, meaning the reported IR of **1.01% remains unchanged** after rounding.
>
> This result demonstrates that the lenient protocol has negligible practical impact on our conclusions.
>
> The lenient protocol was adopted for the following reason: during dataset construction, we identified cases where certain advanced operations (e.g., fillet/chamfer on complex geometry) succeeded under commercial CAD kernels (e.g., Onshape's Parasolid kernel) but failed under OpenCASCADE due to its limited robustness. We retain the lenient criterion to better preserve the fidelity of the original models in the ABC dataset, especially in complex cases. As the experimental results show that this choice does not affect the final performance, to avoid potential misunderstanding, we will include a clarifying explanation of this rationale in the revised manuscript.
>
> ### **(2) R-F1 Drop from SFT to Full Method**
>
> We attribute the R-F1 drop to a reward-driven shift in command composition. Generation commands (extrude, revolve) directly define the primary volumetric structure and are more straightforward to optimize for CD-based rewards, whereas refinement features (fillet, chamfer) contribute more subtly to overall geometry. Consequently, during RL optimization, the model tends to fit the target shape using a greater number of generation commands rather than relying on refinement operations, which also explains the simultaneous increase in G-F1 and the substantial CD improvement.
>
> We thank the reviewer for this suggestion. We will include the discussion in the revised version.
>
> ### **(3) Evaluation Text Descriptions and Human-Authored Prompts**
>
> Evaluation descriptions in the main experiments are generated by Claude-4.5 from ground-truth CAD models, matching the training distribution. We additionally evaluate on **expert-annotated descriptions** from [1], which contains 200 samples, split into two levels: **abstract** (no dimensional information) and **dimension-specified**. Results on the standard subset are as follows:
>
> |Desc. Type |G-F1↑|Median CD↓|Mean CD↓| IR↓|
> |---|---|---|---|---|
> |Abstract |89.41 |77.11 |128.29 |1.00% |
> |Dimension-specified |90.75 |19.37 |85.46 |0.50% |
>
> > R-F1 is not reported here as the models in [1] do not include advanced features
>
> Under dimension-specified descriptions, FutureCAD achieves a median CD of 19.37 and IR of 0.50%, demonstrating strong generalization to human-authored prompts when dimensions are provided. Under abstract descriptions, G-F1 remains high and IR stays at 1.00%, confirming that the model produces valid outputs even without explicit geometric details. The CD increase under abstract prompts is expected, as recovering precise dimensions from abstract descriptions is inherently underdetermined.
>
> > [1] Generating CAD Code with Vision-Language Models for 3D Designs. ICLR 2025
>
> ### **(4) Stronger Baseline**
>
> We adapt CAD-LLaMA to use BRepGround for primitive grounding and evaluate on the Advanced subset:
>
> | G-F1↑ | R-F1↑ | Median CD↓ | Mean CD↓ | IR↓ |
> |---|---|---|---|---|
> | 80.54 | 88.67 | 43.42 | 86.78 | 41.46% |
>
> Replacing the nearest-primitive heuristic with BRepGround yields consistent improvements in all metrics, confirming that grounding quality is an important factor. However, FutureCAD still significantly outperforms the result, especially in IR. This indicates that the performance gain arises from the synergy between our RL optimization and query-based BRepGround.
>
> ### **(5) End-to-End Runtime**
>
> We report the end-to-end runtime breakdown measured on our full test set of **7,168 samples**. All experiments are conducted in a single-process setting without parallelization/batch:
>
> |Stage |Avg. Time per Sample |
> |---|---|
> |LLM program generation (vLLM inference) | 255 ms |
> |CAD kernel execution |125 ms (include avg. 23.6 ms for BRepGround per forward pass) |
> |**Total** |**~380 ms** |
>
> The total latency is suitable for interactive use.
>
> ### **(6) Topology-aware Reward Question**
>
> We thank the reviewer for this insightful suggestion. We agree that incorporating topology-aware rewards is a promising direction.
>
> CAD models are inherently one-to-many: a single target geometry can be constructed through multiple valid sequences, often involving different intermediate topologies. This ambiguity makes the design of topology-aware rewards non-trivial. Therefore, we adopt CD as the primary reward, as it provides a stable signal and directly captures geometric fidelity. We also agree that incorporating topology-aware signals alongside CD could help address the limitations of purely geometry-based rewards, and we consider this a promising direction for future work.

---

> > ### Author Rebuttal · Reviewer_VGxa · 2026-04-03
> >
> > The strict validity check was the most important result - only 2 additional failures out of 7,168 samples under the strict criterion. That puts the main concern to rest. The OpenCASCADE vs Parasolid kernel explanation for keeping the lenient protocol is reasonable.
> >
> > The human-authored prompt evaluation on [1]'s expert annotations was a good addition. Median CD of 19.37 and IR 0.50% under dimension-specified descriptions - that's strong. The CD increase under abstract descriptions is expected; the IR staying at 1.00% is what matters.
> >
> > CAD-LLaMA + BRepGround at 41.46% IR vs 1.01% clearly shows the performance gap comes from the full pipeline, not just the grounding module. The R-F1 drop explanation (RL favoring generation commands over refinement, trading R-F1 for CD improvement) makes sense. Runtime at 380ms is practical.

---

### Official Review · Reviewer_57Xw · 2026-03-13

**Soundness:** 3
**Presentation:** 2
**Significance:** 4
**Originality:** 4
**Overall Recommendation:** 5
**Confidence:** 5

**Summary:**

This paper presents a dataset and method for generating CAD programs from detailed textual descriptions of a part. It generates programs in a variant of CadQuery that has been augmented to support natural-language based queries, using a learned BRepGround model that aligns B-Rep primitive embeddings to textual queries. Generation is performed by a small (7b) LLM that has been tuned by SFT on program description pairs followed by reinforcement learning with Group Sequence Policy Optimization using a chamfer distance reward.

A new construction sequence dataset incorporating advanced (beyond sketch+extrude) features and resolving references was derived from ABC's construction sequences.

**Compliance With Llm Reviewing Policy:**

Affirmed.

**Final Justification:**

This work takes an important step towards automated procedural CAD generation going beyond sketch-and-extrude constructions by evaluating and making interpretable references in the largest and most ubiquitous CAD dataset (ABC). Their generative model produces solid results, and the text-based querying mechanism they developed could be useful in interactive modeling all on its own. I had some minor concerns about missing reproduction details, but these details were provided in the rebuttal. I maintain my recommendation of acceptance.

**Key Questions For Authors:**

Examples of Claude generated text for training the grounding module?

**Limitations:**

yes

**Strengths And Weaknesses:**

Strengths:
- advanced feature dataset derived from ABC: a longstanding weakness of ABC has been that the construction sequences (FeatureScript files) it collected are not interpretable without re-executing them in the developer API due to the references being uninterpretable except at runtime. This paper has done the work of collecting interpretable references. DeepCAD also did this, but stripped out the advanced features
- The extension of CadQuery to support natural language queries has potential applications in interactive CAD editing as well.

Weaknesses:
- missing reproducibility details around LLM steps, particularly templates for the prompts used to generate training data (descriptions of objects and queries)
some reproducibility issues

---

> ### Author Rebuttal · Authors · 2026-03-29
>
> We thank the reviewer for the recognition of the significance and originality of our approach, particularly in addressing the long-standing issue of non-interpretable references in ABC. We appreciate the recognition of our natural language query mechanism in CadQuery as a step toward bridging parametric modeling and B-Rep reasoning for more practical, interactive CAD systems.
>
> **Reproducibility Clarification.**
>
> We thank the reviewer for pointing out the lack of details regarding the prompt design in our data generation pipeline.
>
> We provide the prompt templates below for (1) textual description generation, (2) primitive query generation, and (3) grounding module training queries. We will include the details in the revised version.
>
> **(1) Description generation prompt:**
> ~~~
> You are given an image and the CAD code that generates it. Your task is to write two levels of geometric descriptions:
>
> - **Overall Description (max 60 words)**: Provide a high-level summary of the model's geometry.
> - **Detail Description (max 120 words)**: Provide a precise description of the model's geometry, including dimensions, proportions, and significant geometric features as reflected in the code. If the code contains many parameters, focus on key dimensions such as overall width, height, depth, and counts of repeating features.
>
> Notes:
> - Do NOT describe colors, materials, lighting, rendering effects, or coordinate information.
> - Base your geometric descriptions primarily on the visual information from the image. The code should be used as a reference for dimensions, quantities, or overall structural hints.
>
> The CAD Code:
> {cad_code}
>
> Your answer must be strictly in the following JSON format:
> ```json
> {{
>     "overall_description": "your overall description here",
>     "detail_description": "your detailed description here"
> }}
> ```
> ~~~
>
> **(2) Primitive query generation prompt:**
> ~~~
> You are a CAD B-Rep grounding assistant.
>
> Your task is to describe ONLY the selected geometry (edges and/or faces) in a 3D CAD model using purely geometric language.
> The description will be used for a grounding task: given a B-Rep and text, the model predicts the selected edge/face labels.
>
> ### Inputs
> 1. Image 1 – Original CAD model
> 2. Image 2 – Same model with the selected edges/faces emphasized in red (red highlighting is only a visual cue to identify the selection; do NOT mention color or highlighting in your output)
>
> Additionally provided:
> - Meta information (auxiliary, for reasoning only):
>   - Number of selected edges: {edge_count}
>   - Number of selected faces: {face_count}
>
> The meta information is not required to be explicitly stated in the output. It is only provided to help you infer which geometry is selected.
>
> ### Task
> Describe the selected edges and/or faces ONLY.
> Do NOT describe any attributes that are irrelevant or harmful to the grounding task (e.g., color, material, texture).
>
> ### Description guidelines
> You may describe the selected geometry in **natural, location-focused language**.
> The goal is to provide **sufficient semantic cues to uniquely localize** the selected edges/faces within the entire model.
>
> Descriptions may include, but are not limited to:
> - where the selected geometry is located on the model as a whole
> - how it is positioned relative to the main body or major features
> - whether it lies on an outer boundary, inner region, corner, extremity, or transition area
> - how it relates spatially to nearby prominent structures (e.g., holes, protrusions, recesses)
>
> Use clear, human-like geometric descriptions that help “point to” the selected faces/edges.
> Avoid abstract taxonomies or rigid category listings; focus on **where it is** and **how to find it** in the model.
>
> ### Output format
> Respond in JSON with exactly one key:
> ```json
> {{
>   "descriptions": [
>     "Geometric description from one viewpoint",
>     "Geometric description from a different viewpoint",
>     "Geometric description from another viewpoint"
>   ]
> }}
> ```
> ### Additional requirements
> - Avoid redundant wording.
> - Each description must be **no longer than 100 words**.
> - Use compact, information-dense language.
> ~~~
>
> **(3) Examples of generated queries for grounding (BRepGround):**
> 1. The eight hole rim edges positioned on the upper surface of the part, each marking the circular opening of a cylindrical void that passes completely through the block's thickness.
> 2. The top horizontal edge of the vertical post, where the upper face meets the back face at the highest point of the structure.
> 3. The complete outer boundary surfaces of the cube excluding the top face: comprising the four vertical side walls and the bottom base that collectively form five of the six faces of the rectangular solid.
>
> **Additional reproducibility details.**
> During evaluation, we use a single A100 GPU with vLLM (version 0.10.0) for efficient inference. We set temperature = 0.7 and top\_p = 0.9. For each entry in test set, we sample only one response for all methods to ensure a fair comparison.

---

> > ### Author Rebuttal · Reviewer_57Xw · 2026-04-04
> >
> > Thank you for providing the additional details!

---

### Decision · Program_Chairs · 2026-04-30

**Decision:**

Accept (spotlight)

**Comment:**

The paper initially got high scores. The rebuttal and subsequent discussions clarified some of the initial concerns. All the reviewers liked the work and quickly agreed to accept this work. Congrats.

The paper addresses a relevant problem and shows convincing improvement on the ABC dataset. It is a good step forward.